# Monobodies with potent neutralizing activity against SARS-CoV-2 Delta and other variants of concern

Taishi Kondo[1], Kazuhiro Matsuoka[2], Shun Umemoto[1], Tomoshige Fujino[1], Gosuke Hayashi[1,3], Yasumasa Iwatani[2,4], Hiroshi Murakami[1,5]

**Neutralizing antibodies against the severe acute respiratory syndrome coronavirus 2 (SARS-CoV-2) are useful for patients' treatment of the coronavirus disease 2019 (COVID-19). We report here affinity maturation of monobodies against the SARS-CoV-2 spike protein and their neutralizing activity against SARS-CoV-2 B.1.1 (Pango v.3.1.14) as well as four variants of concern. We selected matured monobodies from libraries with multi-site saturation mutagenesis on the recognition loops through in vitro selection. One clone, the C4-AM2 monobody, showed extremely high affinity ($K_D$ < 0.01 nM) against the receptor-binding domain of the SARS-CoV-2 B.1.1, even in monomer form. Furthermore, the C4-AM2 monobody efficiently neutralized the SARS-CoV-2 B.1.1 ($IC_{50}$ = 46 pM, 0.62 ng/ml), and the Alpha ($IC_{50}$ = 77 pM, 1.0 ng/ml), Beta ($IC_{50}$ = 0.54 nM, 7.2 ng/ml), Gamma ($IC_{50}$ = 0.55 nM, 7.4 ng/ml), and Delta ($IC_{50}$ = 0.59 nM, 8.0 ng/ml) variants. The obtained monobodies would be useful as neutralizing proteins against current and potentially hazardous future SARS-CoV-2 variants.**

## Introduction

The spike protein of severe acute respiratory syndrome coronavirus 2 (SARS-CoV-2) is a trimer that binds the angiotensin-converting enzyme 2 (ACE2) on the host cell surface to initiate infection of the virus (Ke et al, 2020; Wrapp et al, 2020; Yan et al, 2020). The spike protein consists of an S1 subunit containing an $N$-terminal domain, a receptor-binding domain (RBD), and an S2 subunit containing the two heptad repeats, HR1 and HR2. The RBD in the S1 subunit has attracted attention as a possible target domain for neutralizing antibodies, as it is the domain that binds to ACE2.

Various neutralizing antibodies have been developed to target coronavirus disease 2019 (COVID-19) (Chen et al, 2020; Shi et al, 2020; Cao et al, 2020b; Dong et al, 2021; Taylor et al, 2021; Yamin et al, 2021;

Du et al 2021a, 2021b). Some of these neutralizing antibodies have been approved by the United States Food and Drug Administration, and many are in the clinical test stage (Cathcart et al, 2021 *Preprint*; Gottlieb et al, 2021; Gupta et al, 2021; Taylor et al, 2021; Weinreich et al, 2021; Chen et al, 2021a; Du et al, 2021a). Although numerous research studies and clinical tests prove that neutralizing antibodies effectively treat COVID-19 patients, the use of antibodies for common diseases' treatment is challenging because it requires low-cost and large-scale antibody production (Buyel et al, 2017; Corti et al, 2021). To solve this problem, single-domain antibodies (Hamers-Casterman et al, 1993), also called nanobodies that bound to the RBD have been developed (Esparza et al, 2020; Huo et al, 2020; Schoof et al, 2020; Xiang et al, 2020; Güttler et al, 2021; Koenig et al, 2021; Li et al, 2021; Sun et al, 2021; Xu et al, 2021; Ye et al, 2021; Zebardast et al, 2021; Chen et al, 2021b). Nanobodies from a synthetic yeast library have shown moderate neutralizing activity in monomer form (a half-maximal inhibitory concentration; $IC_{50}$ = 6.3 nM) against the Wuhan SARS-CoV-2 pseudovirus and potent neutralizing activity in trimer form ($IC_{50}$ = 120 pM) (Schoof et al, 2020). Other nanobodies obtained by immunization of llamas showed neutralizing activity against the pseudovirus ($IC_{50}$ = 0.3 nM in monomer form, $IC_{50}$ = 0.9 pM in trimer form), although they were produced as an Fc fusion protein by Expi293 cells (Xu et al, 2021). The most potent nanobody obtained recently by immunization of llamas has an $IC_{50}$ = 45 pM in monomer form and an $IC_{50}$ = 1.3 pM in trimer form against the pseudovirus (Xiang et al, 2020).

Non-immunoglobulin–based protein derived from the 10th type III domain of human fibronectin, monobody or adnectin, is one of the smallest back-bone proteins used for in vitro selection and is well expressed in a bacterial system (Koide et al 1998, 2012; Wojcik et al, 2010; Lipovsek 2011). In a previous publication, by using the transcription-translation coupled with puromycin-linker (TRAP) display (Ishizawa et al, 2013; Kondo et al, 2020, 2021a), we developed three monobodies (C4, C6b, and C12b), which simultaneously bind to a distinct epitope in the RBD with nM or sub-nM affinity (Kondo et al, 2020). The C4 and C6b monobodies inhibited the ACE2 and RBD interaction and, in addition, the C6b monobody showed

[1]Department of Biomolecular Engineering, Graduate School of Engineering, Nagoya University, Nagoya, Japan   [2]Department of Infectious Diseases and Immunology, Clinical Research Center, National Hospital Organization Nagoya Medical Center, Nagoya, Japan   [3]Japan Science and Technology Agency (JST), PRESTO, Kawaguchi, Japan   [4]Division of Basic Medicine, Graduate School of Medicine, Nagoya University, Nagoya, Japan   [5]Institute of Nano-Life-Systems, Institutes of Innovation for Future Society, Nagoya University, Nagoya, Japan

Correspondence: iwatani.yasumasa.cp@mail.hosp.go.jp; murah@chembio.nagoya-u.ac.jp

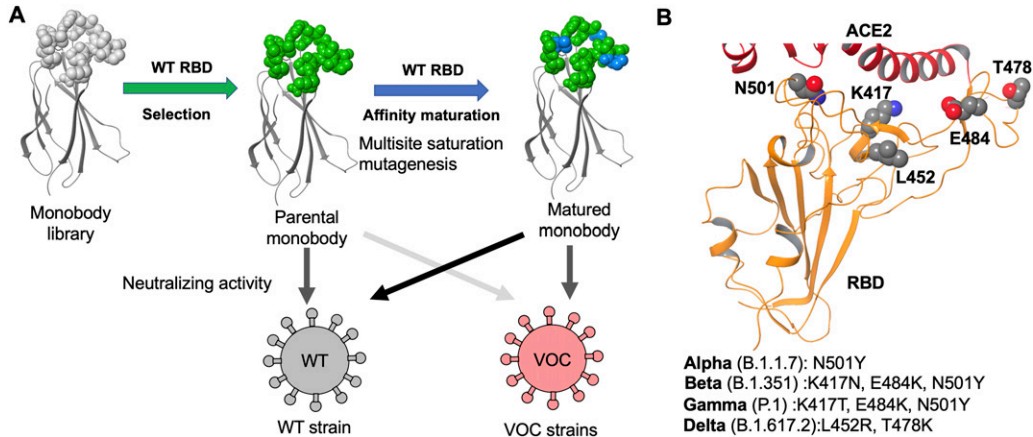

**Figure 1.   Affinity maturation of the monobodies against SARS-CoV-2 wild type and the neutralizing activity of the matured monobody against the wild type and variants of concern (VOCs).**
**(A)** Development of the matured monobody using TRAP display. Multiple saturation mutagenesis was introduced to the BC and FG loops of the parental monobody, and in vitro selection against the SARS-CoV-2 wild-type receptor-binding domain (RBD) was conducted. The obtained matured monobody had enhanced neutralizing activity against SARS-CoV-2 wild type and VOCs. **(B)** Structure of the SARS-CoV-2 RBD (PDB entry 6M0J, Lan et al, 2020). The characteristic residues in the VOCs are labeled as K417, L452, T478, E484, and N501. Abbreviations: SARS-CoV-2, severe acute respiratory syndrome coronavirus 2; RBD, receptor-binding domain; ACE2, angiotensin-converting enzyme 2; WT, wild type; VOCs, variants of concern.

moderate neutralizing activity against the SARS-CoV-2 B.1.1 (Pango v.3.1.14, referred to as the "wild type" in this study) ($IC_{50}$ = 0.5 nM).

This study elucidates optimized sequences of the C4, C6b, and C12b monobodies using the TRAP display from libraries containing multi-site saturation mutagenesis to obtain a matured monobody (Fig 1A). Through two cycles of affinity maturation of the C4 monobody, we successfully obtained the optimized C4-AM2 monobody, which has a high affinity for the wild-type RBD ($K_D$ < 0.01 nM) and potent neutralizing activity against the SARS-CoV-2 wild type ($IC_{50}$ = 46 pM, 0.62 ng/ml) in the monomer form. We also tested the C4-AM2 and C6b monobodies to neutralize four SARS-CoV-2 variants currently of major concern (Fig 1B). The C4-AM2 monobody had high neutralizing activity against the Alpha variant and sub-nM neutralizing activity against the Beta, Gamma, and Delta variants. In contrast, the C6b monobody had nanomolar-level $IC_{50}$ values for all tested SARS-CoV-2 variants.

## Results

### Affinity maturation of C4 monobody

In previous research, we obtained three monobodies from an initial synthetic library and used them to detect or neutralize the SARS-CoV-2 wild type as they already had sub-nM to nM affinities against the RBD (Kondo et al, 2020). In this study, we first evaluated the importance of each residue in the BC and FG loops of the C4 monobody. We prepared four libraries from the C4 monobody by adding multi-site saturation mutagenesis of up to six residues in the BC and FG loops (Fig 2A, upper panel). The theoretical diversity of a library with six random residues is $1 \times 10^9$ (NNK$_6$; N = A, C, G, T; K = G or T; $32^6$). Therefore, it was easily covered by the diversity of the in vitro selection system (0.1 μM monobody–mRNA complexes in 5 μl reaction mixture; $3 \times 10^{11}$ molecules). After three rounds of selection

against the RBD of the SARS-CoV-2 wild type (Fig 2B), the obtained cDNA sequences were analyzed by next-generation sequencing, and WebLogo (Crooks et al, 2004) was used to display the probability of the amino acids in each position. The result indicated the amino acids in the 10 positions in the loops of the C4 monobody (the BC loop, PxxxxxxYQz; the FG loop, WTGxxPWzWxxN) were important for binding to its target (Fig 2A, upper panel). The two positions indicated by "z" were conserved aromatic residues, and some positions have mid to weak preferences for particular amino acids. Interestingly, the probability at the third and fourth positions in the BC loop of the C4 monobody suggested that Gly³Gly⁴ were preferable than the original residues (Ser³Arg⁴). Therefore, we prepared the C4-AM1 monobody to study the effect of these different mutations.

During the optimization of the monobody expression, we found that the extension of the natural sequence at the C-terminus improved its solubility (Fig S1; C4-AM2 monobody, vide infra). Because such monobodies did not have a Cys residue in their body sequence, we added a Cys residue at the C-terminus of the C4-AM1 monobody (Fig S2A and B). We then modified it with a maleimide-biotin reagent. The resulting monobody was immobilized on a sensor chip, and solutions with various RBD concentrations were used for affinity measurements by biolayer interferometry (BLI). As expected, the $K_D$ value was improved from 0.96 to 0.11 nM (Fig 2C and Table 1).

For the second cycle of affinity maturation, we prepared a library created from the C4-AM1 monobody by adding multiple semi-saturation mutagenesis to the 10 residues in the BC and FG loops (Fig 2A), resulting in a library with theoretical diversity ($5 \times 10^9$). After 10 rounds of selection against the RBD with extensive washing steps (Fig 2B), the obtained cDNA sequences were analyzed by next-generation sequencing. The probability of the amino acids in each position indicated that the mutation from Phe to Trp at the BC loop's last position could improve the monobody's affinity (Fig 2A). Accordingly, the most abundant clone, the C4-AM2

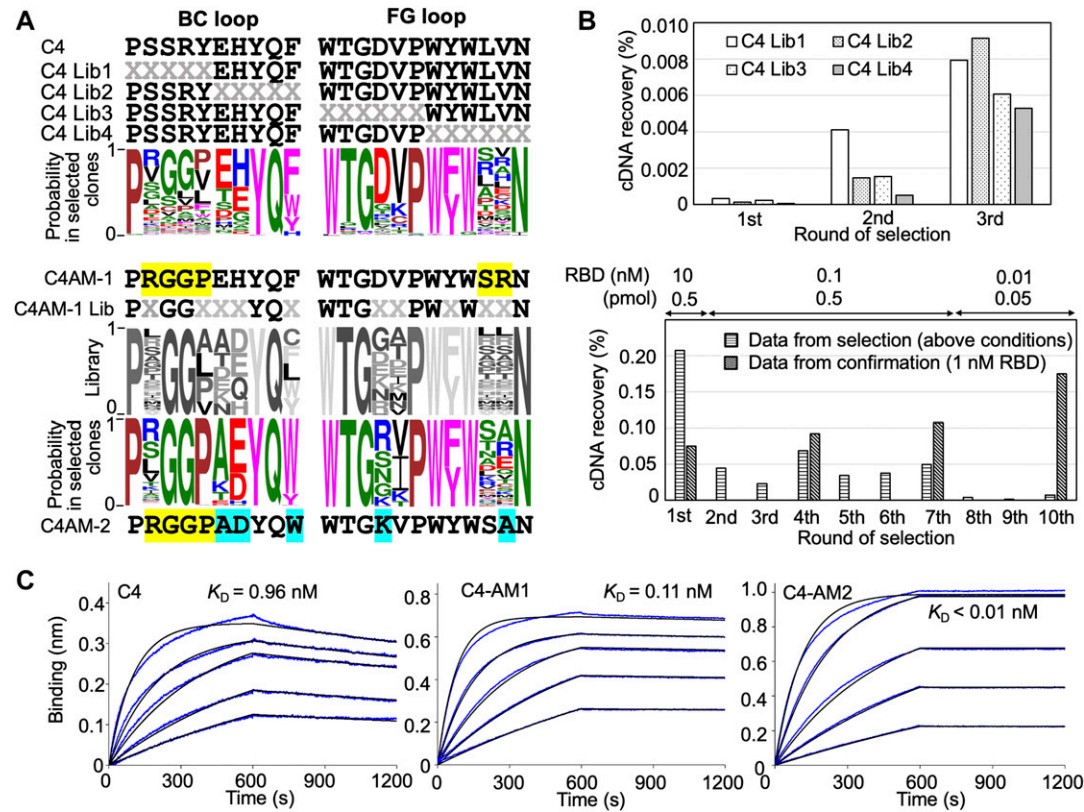

**Figure 2. Affinity maturation of the C4 monobody.**
**(A)** Sequences of the BC and FG loops in the libraries and selected monobodies. Saturation mutagenesis (X) was introduced by NNK codons (N = A, C, G, T; K = G or T; 32 codons/20 aa) at five to six constitutive resides in the BC and FG loops. The probability of amino acids at each position in the loops of the selected clones was shown by WebLogo. The mutated residues in the matured C4-AM1 monobody are highlighted in yellow. The second library was designed to enhance the activity of the C4-AM1 monobody. Partial saturation mutagenesis (X) shown in the WebLogo (gray) was introduced into the library. The mutated residues in the matured C4-AM2 monobody are highlighted in cyan. **(B)** Selection progress by the TRAP display. A 1 nM receptor-binding domain (RBD) (0.1 pmol) concentration was used for the C4 Lib1-4 selection. The concentration of RBD is stated in the figure used for the C4-AM1 Lib selection. A confirmation experiment (1 nM RBD, 0.5 pmol) was also performed at the 1st, 4th, 7th, and 10th rounds to observe an enrichment of active monobodies. **(C)** Determination of kinetic parameters by BLI. A biotin-labeled monobody was immobilized on a streptavidin-sensor chip, and SARS-CoV-2 RBD (2.5, 5, 10, 20, and 40 nM) was used in the kinetic analysis. The data are depicted in blue, and the fitted 1:1 binding model in black. The determined kinetic parameters of the monobodies are provided in Table 1. Abbreviations: BLI, Bio-layer interferometry; Lib, library. Other abbreviations are as for Fig 1.

**Table 1. Kinetic parameters of monobodies against the SARS-CoV-2 wild-type receptor-binding domain determined by BLI.**

| Names | BC loop | FG loop | $K_D$ (nM) | $k_{on}$ (1/Ms) × 10$^5$ | $k_{off}$ (1/s) × 10$^{-4}$ |
|---|---|---|---|---|---|
| C4 | PSSRYEHYQF | WTGDVPWYWLVN | 0.96 | 2.4 | 2.3 |
| C4-AM1 | PRGGPEHYQF | WTGDVPWYWSRN | 0.11 | 3.3 | 0.38 |
| C4-AM2 | PRGGPADYQW | WTGKVPWYWSAN | <0.01 | 2.3 | <0.01 |
| C6b | GGDYVGYY | TYNGPWIYGYEEI | 0.51 | 1.2 | 0.63 |
| C6b-AM1 | GGAGAHLY | TYNGPWIYGYEEI | 1.7 | 1.2 | 2.0 |
| C6b-GS | GSGSGS | TYNGPWIYGYEEI | 4.3 | 1.3 | 5.6 |
| C6b-PAVT | PAVT | TYNGPWIYGYEEI | 7.2 | 1.6 | 11 |
| C6b-AM2[a] | WIMQLDSGYWDR | TYNGPWIYGYEEI | 0.36 | 0.95 | 0.34 |
| C12b | EIYYEIGD | RLWGYYTQWD | 0.67 | 1.7 | 1.1 |
| C12b-AM1 | GLGSSFGD | RLWGYYTQWD | 1.0 | 1.9 | 1.9 |
| C12b-AM2[a] | MHWYDQGDTS | RLWGYYTQWD | <0.01 | 1.6 | <0.01 |

[a]The C6b-AM2 and C12b-AM2 monobodies had two amino acids (VR) deletion after the BC loop.

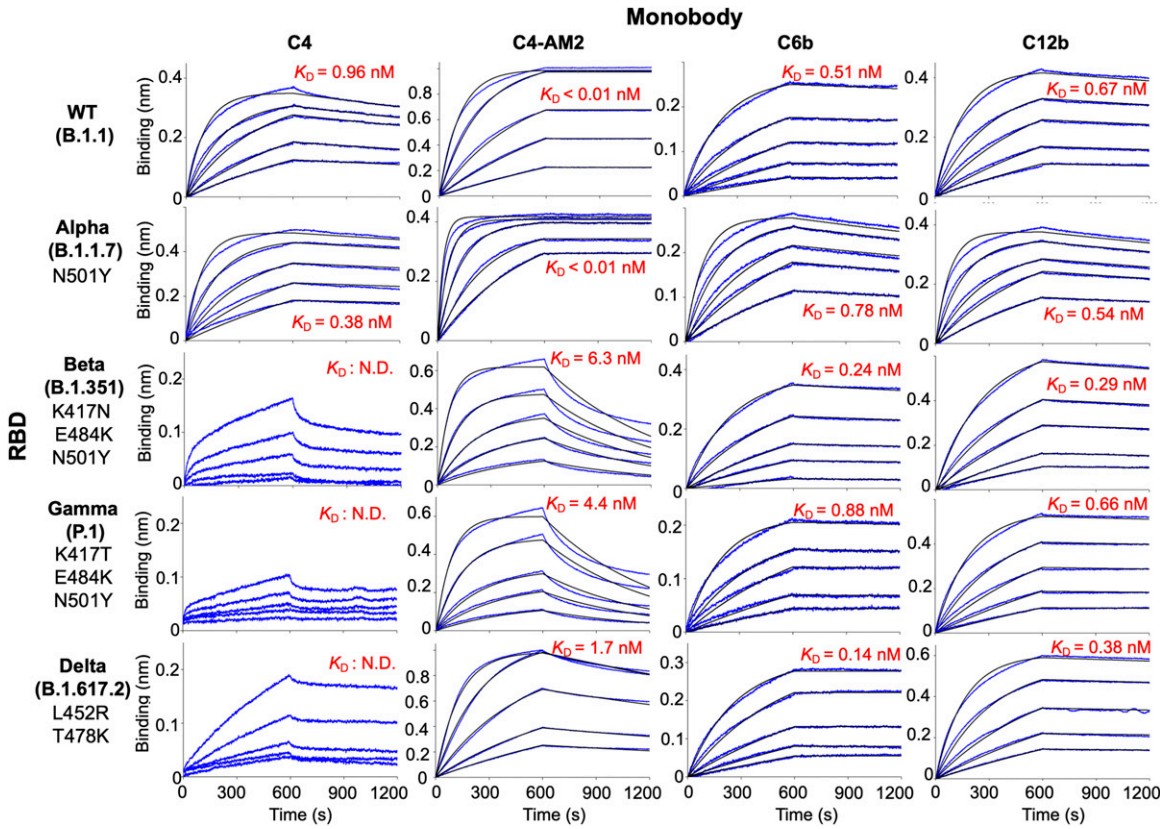

**Figure 3. Binding activity of monobodies against the receptor-binding domains of SARS-CoV-2 wild type and variants of concern analyzed by BLI.**
Biotin-labeled monobody was immobilized on a streptavidin-sensor chip, and the receptor-binding domain of SARS-CoV-2 wild type, Alpha, Beta, Gamma, and Delta (2.5, 5, 10, 20, and 40 nM) variants were used in the kinetic analysis. The data are depicted in blue, and the fitted 1:1 binding model in black. The determined kinetic parameters of the monobodies are provided in Tables 1 and 2. The data from Fig 2 (C4, C4-AM2), S3 (C6b), and S4 (C12b) were included to show the complete matrix. Abbreviations are as mentioned previously.

monobody, showed unmeasurable $k_{off}$ values, demonstrating an ultra-high affinity toward the RBD ($K_D < 0.01$ nM) (Fig 2C and Table 1), even in the monomer form.

### Affinity maturation of C6b and C12b monobodies

We also evaluated the importance of each residue in the BC and FG loops of the C6b monobody (Fig S3A and B). We prepared three kinds of libraries from the C6b monobody and conducted affinity maturation selection as previously described. The results indicated that almost all residues in the FG loop (TYNGPWIYGYEEI) of the C6b monobody were important for binding to its target. In contrast, the BC loop contributed less to the binding. Interestingly, the replacement of the C6b monobody's BC loop with the original fibronectin sequence (PAVT) or a flexible linker (GSGSGS) increased the $K_D$ values up to 14 times (Fig S3C and Table 1; $K_D = 7.2$ nM for PAVT and $K_D = 4.3$ nM for GSGSGS), suggesting that the BC loop contributes to the binding. Because the additional affinity maturation using the BC loop randomized library did not significantly improve the $K_D$ value (Fig S3C; C6b-AM2, $K_D = 0.36$ nM), we decided to use the C6b monobody for further studies.

A similar experiment was performed for the C12b monobody (Fig S4A and B). The results indicated that the amino acids in the $Gly^7$ of

the BC loop and the FG (RLWGYYTQWD) loop were important for binding to the RBD. Further affinity maturation using the BC loop randomized library provided the high-affinity C12b-AM2 monobody ($K_D < 0.01$ nM) (Fig S4B and C and Table 1).

### The binding affinity of monobodies to the RBDs of SARS-CoV-2 variants of concern (VOCs)

Emerging SARS-CoV-2 variants are causing serious problems globally as some have higher infection efficiency and an ability to escape neutralizing antibodies (Garcia-Beltran et al, 2021; Harvey et al, 2021). We aimed to determine if the monobodies could bind to all four VOCs. The BLI analysis demonstrated that all of the monobodies maintained their ability to bind to the RBD of the Alpha variant (B.1.1.7; Fig 3, second row; Table 2; $K_D = 0.38$ nM for C4, $K_D < 0.01$ nM for C4-AM2, 0.78 nM for C6b, and 0.54 nM for C12b). The C6b and C12b monobodies also maintained the $K_D$ values for the Beta (B.1.351), Gamma (P.1), and Delta (B.1.617.2) variants (Fig 3, third and fourth, columns; Table 2) ($K_D = 0.24$ nM [Beta], 0.88 nM [Gamma], 0.14 nM [Delta] for C6b; $K_D = 0.29$ nM [Beta], 0.66 nM [Gamma], and 0.38 nM [Delta] for C12b), whereas affinities of the C4 and C4-AM2 monobodies were diminished toward these variants (Fig 3, first and second columns; Table 2) ($K_D$ was not determined towards the

**Table 2. Kinetic parameters of monobodies against the SARS-CoV-2 wild type and variants of concern receptor-binding domain determined by BLI.**

| Receptor-binding domain | Parameters | C4 | C4-AM2 | C6b | C12b |
|---|---|---|---|---|---|
| WT (B.1.1) | $K_D$ (nM) | 0.96 | <0.01 | 0.51 | 0.67 |
| | $k_{on}$ (1/Ms) × $10^5$ | 2.4 | 2.3 | 1.2 | 1.7 |
| | $k_{off}$ (1/s) × $10^{-4}$ | 2.3 | <0.01 | 0.63 | 1.1 |
| Alpha (B.1.1.7) | $K_D$ (nM) | 0.38 | <0.01 | 0.78 | 0.54 |
| | $k_{on}$ (1/Ms) × $10^5$ | 2.8 | 7.8 | 2.5 | 3.2 |
| | $k_{off}$ (1/s) × $10^{-4}$ | 1.1 | <0.01 | 1.9 | 1.7 |
| Beta (B.1.351) | $K_D$ (nM) | N.D. | 6.3 | 0.24 | 0.29 |
| | $k_{on}$ (1/Ms) × $10^5$ | N.D. | 2.6 | 1.3 | 1.3 |
| | $k_{off}$ (1/s) × $10^{-4}$ | N.D. | 16 | 0.31 | 0.39 |
| Gamma (P.1) | $K_D$ (nM) | N.D. | 4.4 | 0.88 | 0.66 |
| | $k_{on}$ (1/Ms) × $10^5$ | N.D. | 3.4 | 1.0 | 0.99 |
| | $k_{off}$ (1/s) × $10^{-4}$ | N.D. | 15 | 0.89 | 0.65 |
| Delta (B.1.617.2) | $K_D$ (nM) | N.D. | 1.7 | 0.14 | 0.38 |
| | $k_{on}$ (1/Ms) × $10^5$ | N.D. | 1.9 | 0.76 | 1.4 |
| | $k_{off}$ (1/s) × $10^{-4}$ | N.D. | 3.2 | 0.11 | 0.52 |

The data for the wild type were the same as in Table 1. N.D., not determined.

Beta, Gamma, and Delta variants for C4; $K_D$ = 6.3 nM [Beta], 4.4 nM [Gamma], 1.7 nM [Delta] for C4-AM2). The dissociation curves for the C4-AM2 monobody against the RBD of the Beta, Gamma, and Delta variants were not matched to the fitting curves using a 1:1 binding model, which indicated that there might be two dissociation rate constants (Fig S5 and Table S1).

Because the RBD's N501Y mutation was seen in the Alpha variant, some of the other three mutations (K417T/N, L452R, and E484K) could cause reduced RBD affinity of the C4 and C4-AM2 monobodies. To elucidate the effect of each mutation on the RBD and monobody binding, we analyzed the binding of the monobodies with RBDs containing point mutations (Fig S6). The results suggested that the E484K and L452R mutations were the primary cause of the reduced RBD affinities of the C4 and C4-AM2 monobodies, whereas the K417T mutation had no influence.

## Neutralization activities of monobodies against the SARS-CoV-2 wild type

Next, we examined the neutralizing activity of the C4, C4-AM2, and C6b monobodies against the B.1.1 lineage isolate of SARS-CoV-2 (GISAID ID# EPI_ISL_568558) using the modified neutralization assay described previously (Kondo et al, 2020). Briefly, we incubated serially diluted monobody solutions with live virus (200 TCID$_{50}$/well) for 1 h at 37°C in 96-well culture plates, followed by the infection of VeroE6/TMPRSS2 cells (Matsuyama et al, 2020) for 1 h at 37°C. After an 18-h incubation with fresh medium, the SARS-CoV-2 mRNA levels in the supernatant were measured by RT-qPCR. As a negative control, we used the C12b monobody, which bound to the RBD without inhibiting the ACE2/RBD interaction and had no neutralizing activity as previously shown (Fig 4; B.1.1, red line). As a positive control, we used the commercially available AM-128 neutralizing antibody, a chimeric monoclonal antibody that combines the constant domains of the human IgG molecule with a mouse variable region. The $IC_{50}$ value (0.11 nM; Fig 4; B.1.1, ocher line; Table 3) of the control AM-128 neutralizing antibody was comparable to the manufacturing company's data ($IC_{50}$ = 0.03 nM against the wild-type SARS-CoV-2 pseudovirus). The C6b monobody's $IC_{50}$ value ($IC_{50}$ = 1.6 nM; Fig 4; B.1.1, green line; Table 3) was slightly increased from that previously reported ($IC_{50}$ = 0.5 nM). This increase may have resulted from the reduced steric hindrance of the RBD/ACE2 interaction as the Nus-Tag fusion protein (60 kD) from the previous construct was removed to prepare the monobody without a fusion protein (13 kD). More importantly, the C4-AM2 monobody showed ultra-potent neutralizing activity ($IC_{50}$ = 46 pM; Fig 4, B.1.1, black line; Table 3), whose $IC_{50}$ was more than 100 times lower than the C4 monobody ($IC_{50}$ = 6.6 nM; Fig 4, B.1.1, blue line; Table 3). Moreover, the $IC_{50}$ in mass concentration was 27 times lower than the control neutralizing antibody AM-128 (Table 3; 0.62 ng/ml versus 17 ng/ml) because of the low molecular weight of the monobody. To our knowledge, the C4-AM2 monobody is one of the most potent neutralizing proteins against the SARS-CoV-2 wild type in monomer form obtained by immunization and selection methods or by de novo design (Xiang et al, 2020; Cao et al, 2020a).

## Neutralizing activities of monobodies against the SARS-CoV-2 VOCs

Next, we tested the monobodies to neutralize the emerging SARS-CoV-2 variants, Alpha, Beta, Gamma, and Delta. The C6b monobody maintained its $IC_{50}$ values among all the variants ($IC_{50}$ = 1.6–8.4 nM; Fig 4 and Table 3), consistent with the previously reported $K_D$ values against the mutant RBDs ($K_D$ = 0.24–0.88 nM; Table 2). This result indicated the uniform usability of the C6b monobody for neutralization of various SARS-CoV-2 variants.

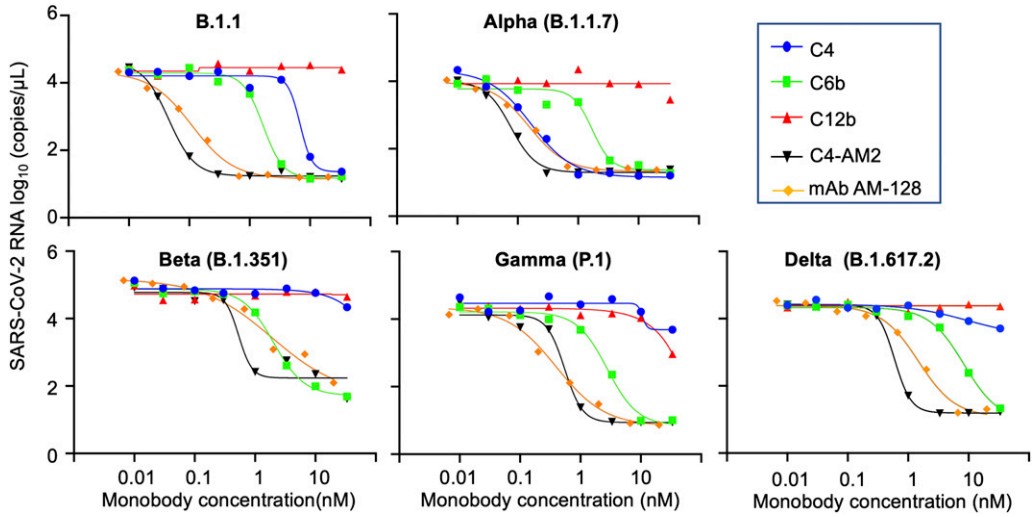

**Figure 4. Monobody/antibody-mediated neutralization of SARS-CoV-2 wild type and variants of concern infection in VeroE6/TMPRSS cells.**
The x-axis value indicates the final concentration of the monobodies (C4, C6b, C12b, and C4-AM2) or the SARS-CoV-2 neutralizing monoclonal antibody, AM-128, for each assay well. The y-axis value displays $\log_{10}$ of each viral RNA copy number in the supernatant. Data are represented as the geometric mean of three independent assays. Abbreviations are as mentioned previously.

**Table 3. Half maximal inhibitory concentration of monobodies and mAb AM-128 against SARS-CoV-2 wild type and variants of concern.**

| | | C4 | C4-AM2 | C6b | mAb AM-128 |
|---|---|---|---|---|---|
| WT (B.1.1) | $IC_{50}$ (nM) | 6.6 | 0.046 | 1.6 | 0.11 |
| | $IC_{50}$ (ng/ml) | 88 | 0.62 | 22 | 17 |
| Alpha (B.1.1.7) | $IC_{50}$ (nM) | 0.17 | 0.077 | 1.6 | 0.15 |
| | $IC_{50}$ (ng/ml) | 2.2 | 1.0 | 22 | 23 |
| Beta (B.1.351) | $IC_{50}$ (nM) | N.D. | 0.54 | 1.9 | 0.40 |
| | $IC_{50}$ (ng/ml) | N.D. | 7.2 | 25 | 60 |
| Gamma (P.1) | $IC_{50}$ (nM) | N.D. | 0.55 | 2.6 | 1.9 |
| | $IC_{50}$ (ng/ml) | N.D. | 7.4 | 34 | 285 |
| Delta (B.1.617.2) | $IC_{50}$ (nM) | N.D. | 0.59 | 8.4 | 1.5 |
| | $IC_{50}$ (ng/ml) | N.D. | 8.0 | 114 | 225 |

N.D., not determined.

Conversely, the C4 monobody was more effective against the Alpha variant ($IC_{50}$ = 0.17 nM; Fig 4 and Table 3) and lost its activity against the Beta, Gamma, and Delta variants ($IC_{50}$ was not determined; Fig 4). These results were partially consistent with the $K_D$ values observed for the variants' RBDs, although the $IC_{50}$ values of the Alpha variant were slightly lower than expected.

The C4-AM2 monobody maintained its $IC_{50}$ values against the Alpha variant ($IC_{50}$ = 77 pM; Fig 4 and Table 3), which was consistent with the affinity measurement (Fig 3). Surprisingly, the C4-AM2 monobody showed sub-nM $IC_{50}$ values against the Beta, Gamma, and Delta variants ($IC_{50}$ = 0.54 nM [Beta], 0.55 nM [Gamma], 0.59 nM [Delta]; Fig 4 and Table 3). Because the $K_D$ values observed against these variants' RBDs were 1.7–6.3 nM, the $IC_{50}$ values were 10 times lower than expected, suggesting that the C4-AM2 monobody may have two dissociation rates (Fig S5), and the slow dissociation rate might partially contribute to the low $IC_{50}$ values.

## Discussion

In this study, we performed affinity maturation of monobodies determined as SARS-CoV-2 RBD binders in our previous research. Through two cycles of affinity maturation using the TRAP display, we obtained the C4-AM2 monobody, which has enhanced affinity from the C4 monobody's $K_D$ = 0.96 nM to $K_D$ < 0.01 nM. The C4-AM2 monobody's $k_{off}$ value could not be determined because of the extremely slow dissociation rate. The C4-AM2 monobody also had a very low $IC_{50}$ value against the SARS-CoV-2 wild type ($IC_{50}$ = 0.62 ng/ml), which was 27 times better in mass concentration than those of the control antibody (mAb AM-128; $IC_{50}$ = 17 ng/ml). This low $IC_{50}$ value is comparable in molar concentration of those of the antibodies reported as being highly active for neutralizing the SARS-CoV-2 pseudovirus (casirivimab, $IC_{50}$ = 50 pM, 7.5 ng/ml; imdevimab $IC_{50}$ = 27 pM, 4.1 ng/ml), and thus better in mass concentration (US Food and Drug Administration, 2021).

We also found that the affinities of the C4 and C4-AM2 monobodies against the Alpha variant's RBD stayed in a similar range of values ($K_D$ < 0.01 nM) but was diminished against those of the Beta ($K_D$ = 6.3 nM), Gamma ($K_D$ = 4.4 nM), and Delta ($K_D$ = 1.7 nM) variants. In contrast, the C6b and C12b monobodies showed similar affinity against all four variants ($K_D$ = 0.14–0.88 nM for C6b, $K_D$ = 0.29–0.66 nM for C12b). Surprisingly, in spite of the diminished $K_D$ values of the C4-AM2 monobody, it showed a sub-nM $IC_{50}$ value against the SARS-CoV-2 Beta ($IC_{50}$ = 0.54 nM), Gamma ($IC_{50}$ = 0.55 nM), and Delta ($IC_{50}$ = 0.59 nM) variant infection. The C6b and C4-AM2 monobodies could be useful for the rapid treatment of patients infected by the SARS-CoV-2's VOCs and potential new hazardous variants in the future.

Multimerization of the C4-AM2 or C6b monobodies could be a solution to improve $IC_{50}$ values, as dimerization and trimerization of a nanobody could improve the $IC_{50}$ values between 2 and 200 times against the SARS-CoV-2 wild type or pseudovirus (Huo et al, 2020; Schoof et al, 2020; Xiang et al, 2020; Chen et al, 2021b; Güttler et al,

2021; Koenig et al, 2021; Li et al, 2021; Xu et al, 2021; Zebardast et al, 2021). We know from previous research (Kondo et al, 2020) that simple dimerization is insufficient to improve monobody activity. Therefore, careful design of the multimerization strategy would be required to improve the monobodies $IC_{50}$ values in future work.

# Materials and Methods

### Materials

The oligonucleotides and synthetic DNA templates were purchased from either Fasmac Co., Ltd., Nippon Bio Service, or Integrated DNA Technologies. The sequences were listed in Supplemental Data 1. The N2 primer/probe set for RT-qPCR was purchased from Thermo Fisher Scientific. Biotinylated SARS-CoV-2 S protein RBD His, Avitag SPD-C82E9 (WT), SPD-C82E7 (K417T, E484K, and N501Y), RBD His Tag SPD-S52H6 (WT), SPD-C52Hn (N501Y), SPD-C52Hp (K417N, E484K, and N501Y), SRD-C52H3 (E484K), SPD-C52Ht (K417T), SPD-C52Hh (L452R and T478K), SPD-C52He (L452R), and SPD-C52Hr (K417T, E484K, and N501Y) were all purchased from ACROBiosystems. The restriction enzymes were obtained from New England Biolabs. The preparation of the cell-free translation system, *Pfu-S* DNA polymerase, and Moloney murine leukemia virus reverse transcriptase (MMLV) have been described in previous reports (Shimizu et al, 2001; Ohashi et al, 2007; Reid et al, 2012; Kondo et al, 2020).

### Preparation of monobody mRNA libraries

To prepare the DNA libraries for affinity maturation experiments, FN3F-QAN.F1 (0.5 $\mu$M), ssDNA-FN3F.F3 (0.5 $\mu$M), FN3F.F5 (0.5 $\mu$M), and the BC-L.F2 (0.5 $\mu$M) and FG-L.F4 (0.5 $\mu$M) were ligated with T4 DNA ligase (2.5 $\mu$l in total, 1.25 pmol for each oligonucleotide) with the assistance of MonoE1c1NH2-NNN.R25 (1 $\mu$M), MonoE1c2-NNN1-NH2.24 (1 $\mu$M), MonoE1c3NH2.R19 (1 $\mu$M), and MonoE1c4NH2-NNN.R24 (1 $\mu$M) (see Table S2 for the combination of BC-L.F2 and FG-L.F4 [0.5 $\mu$M] oligonucleotides used in the preparation of each library). After ligation, the mixture was added to the reaction mixture (1.2 ml in total; 10 mM Tris–HCl, pH 8.4, 100 mM KCl, 0.1% [vol/vol] Triton X-100, 2% [vol/vol] DMSO, 2 mM MgSO₄, 0.2 mM each dNTP, 0.375 $\mu$M T7SD8M2.F44, 0.375 $\mu$M G5S-4Gan21-3.R42, and 2 nM of *Pfu-S* DNA polymerase). The DNA libraries were amplified by eight cycles of PCR. The mRNA libraries were prepared by in vitro run-off transcription under the following conditions: 40 mM Tris–HCl, pH 8.0, 1 mM spermidine, 0.01% (vol/vol) Triton X-100, 10 mM DTT, 30 mM MgCl₂, 5 mM of each NTP, the DNA library, and 0.18 $\mu$M T7 RNA polymerase. The synthesized mRNA was purified by phenol/chloroform extraction and isopropanol precipitation. The mRNA concentration was determined by OD at 260 nm. The mRNA/HEX-mPuL was prepared by annealing HEX-mPuL (5 $\mu$M) and mRNA (8.3 $\mu$M) in annealing buffer (25 mM HEPES-K, pH 7.8, 200 mM potassium acetate) by heating the solution (10 $\mu$l) to 95°C for 2 min and cooling to 25°C. The resulting complex was used directly in the first-round of selection.

The C6-AM2 and C12-AM2 DNA library was prepared using a similar procedure of previously published articles (Kondo et al

2020, 2021b). To prepare A-fragment DNA from the monobody library, FN3F-QAN.F1 (1 $\mu$M), FN3F.F3short39 (1 $\mu$M), and MonobodyHL1Co10 (0.5 $\mu$M) or MonobodyHL1Co12 (0.5 $\mu$M) were ligated by T4 DNA ligase (150 $\mu$l in total, 150 pmol for each oligonucleotide) with the assistance of MonoE1c1NH2-NNN.R25 (2 $\mu$M), and MonoE1c2-NNN1-NH2.24 (2 $\mu$M). As codons for the randomized residues, we used a codon mix with the following ratios: 20% Tyr, 10% Ser, 15% Gly, 10% Trp, and 3% each of the other amino acids (except Cys) which is similar to the original cocktail (30% Tyr, 15% Ser, 10% Gly, 5% Phe, 5% Trp, and 2.5% each of the other amino acids [except for Cys]) (Wojcik et al, 2010; Koide et al, 2012). After ligation, the DNA libraries were amplified using T7SD8M2.F44, FN3F1-2-3-GSBsaI.R34, and *Pfu-S* DNA polymerase (30 ml in total, six cycles of PCR). The B-fragment DNA was prepared via the same procedure and using ssDNA-FN3F.F3, FN3F.F5, C6L2-3.F56/C12L2-3.F47, MonoE1c3NH2.R19, and MonoE1c4NH2-NNN.R24 for ligation, and FN3F3-4-5-GSBsaI.F34 and G5S-4Gan21-3.R42 for amplification.

The amplified BC and FG fragment DNAs were purified by phenol/chloroform extraction and isopropanol precipitation. One end of each DNA product was digested with *BsaI* (New England Biolabs) as per the manufacturer's protocol, and the DNA products were then purified by phenol/chloroform extraction and isopropanol precipitation. The products were ligated to each other (0.15 $\mu$M, 200 $\mu$l) to synthesize full-length DNA products. The DNA libraries were amplified using T7SD8M2.F44, G5S-4Gan21-3.R42, and *Pfu-S* DNA polymerase (40 ml in total, five cycles of PCR). The mRNA/HEX-mPuL was prepared through a similar procedure to that described above. The resulting complex was used directly in the first-round of selection.

### In vitro selection of monobodies against SARS-CoV-2 spike protein RBD by the TRAP display

For the first round of selection, 1 $\mu$M mRNA/HEX-mPuL was added to a reconstituted translation system, and the reaction mixture (5 $\mu$l) was incubated at 37°C for 30 min. After the reaction, 1 $\mu$l of 100 mM EDTA (pH 8.0) was added to the translation mixture. Reverse transcription mixture (3 $\mu$l; 150 mM Tris–HCl, pH 8.4, 225 mM KCl, 75 mM MgCl₂, 16 mM DTT, 1.5 mM each dNTP, 7.5 $\mu$M FN3S.R29 primer, and 3.4 $\mu$M MMLV) was added to the translation mixture, and the resulting solution was incubated at 42°C for 15 min. The buffer was changed to HBST buffer (50 mM Hepes-K, pH 7.5, 300 mM NaCl, and 0.05% [vol/vol] Tween 20) using Zeba Spin Desalting Columns. To remove the bead binders, the resulting solution was mixed with 4 $\mu$l of Dynabeads M-280/M-270 streptavidin (1:1) (Thermo Fisher Scientific) at 25°C for 10 min. This negative selection step was repeated another two times. The supernatant was diluted with the HBST buffer (90 $\mu$l) and mixed with 1 $\mu$l of 100 nM biotinylated SARS-CoV-2 RBD-Avitag (1 nM, final concentration); the resulting solution was incubated at 25°C for 3 min. The target proteins were collected by mixing with 1 $\mu$l of Dynabeads M-270 streptavidin for 1 min. The collected beads were washed with 50 $\mu$l of the HBST buffer for 1 min three times, and 100 $\mu$l of PCR premix (1 ml of 10 mM Tris–HCl, pH 8.4, 50 mM KCl, 0.1% [vol/vol] Triton X-100, 2 mM MgCl₂, and 0.25 mM each dNTP) was added. The beads were heated at 95°C for 5 min, and the amount of eluted cDNA was quantified by SYBR green-based

quantitative PCR using T7SD8M2.F44 and FN3Lip.R20 as primers. The eluted cDNA was PCR-amplified using T7SD8M2.F44, G5S-4Gan21-3.R42, and *Pfu-S* DNA polymerase and purified by phenol/chloroform extraction and isopropanol precipitation. From the following selection, the resulting DNA (about 3–10 nM final concentration) was added to the TRAP system, and the reaction mixture (5 $\mu$l) was incubated at 37°C for 30 min. The other procedure was similar to the above description. After the final round of selection, the sequences of the recovered DNA were analyzed using an Ion Torrent instrument (Thermo Fisher Scientific). The conditions of the second cycle selections are listed in Tables S3 and S4.

### Construction of expression vectors of monobodies

For monobody expression, pQ107-S1-MonoS-C4 was generated by cloning DNA I (Supplemental Data 1) into the *Nde*I-*Hind*III site of pQCSoHis-FN3FB1Xb. Each cloned DNA insert was amplified using the primer sets (Table S5) and templates (DNA fragment II) with *Pfu-S* DNA polymerase. The resulting DNA was further amplified using Q107delMonoS-ex.F71 and Q107EIDKPSQC-ex.R71, followed by the use of Q107.F40 and Q106.R40. The PCR products were cloned into the *Bam*HI-*Hind*III site of pQ107-S1-MonoS-C4 using the patch cloning method (Taniguchi et al, 2013) to generate pQ107-S1-MonoS-X (where X represents the clone name). Plasmids [X = C4-AM2-(−), C4-AM2-(EIDK), C4-AM2-(EIDKPSQ)] were prepared using Q107-MonoS.R67, Q107-MonoS-EIDK.R67, and Q107-MonoS-EIDKPSQ.R67 instead of Q107EIDKPSQC-ex.R71.

### Expression and purification of monobodies

The expression vector was transformed into *Escherichia coli* BL21(DE3)pLysS, and the clone was grown on an LB plate with 100 $\mu$g/ml ampicillin, 20 $\mu$g/ml chloramphenicol, and 2% (wt/vol) glucose. The resulting *E. coli* were inoculated into 3 ml of LB with 100 $\mu$g/ml ampicillin, 20 $\mu$g/ml chloramphenicol, and 5% (wt/vol) glucose, then grown at 37°C. The resulting overnight culture was added to 300 ml of LB with 100 $\mu$g/ml ampicillin, 20 $\mu$g/ml chloramphenicol, and 5% (wt/vol) glucose and grown at 37°C. After reaching an $A_{600}$ value of 0.5, protein expression was induced with 0.5 mM of IPTG for 24 h at 25°C. The cells were pelleted and resuspended in lysis buffer (20 mM Hepes-K, pH 7.8, 10 mM imidazole, pH 7.8, 300 mM KOAc, 1 M KCl, 0.2 mM DTT, and 1 mM PMSF) and lysed using a sonicator. After adding 10% (vol/vol) glycerol, the lysate was clarified by centrifugation (15,300*g*, 30 min, 4°C) followed by filtration. For affinity purification, an IMAC column (Bio-Rad) was connected to an NGC chromatography system (Bio-Rad) equilibrated with buffer A (20 mM Hepes-K, pH 7.8, 10 mM imidazole, 300 mM KOAc, 1 M KCl, 0.2 mM DTT, 10 mM MgSO$_4$, 2 mM ATP, 10% [vol/vol] glycerol). After loading the lysate, the column was washed with buffer A followed by buffer B (20 mM Hepes-K, pH 7.8, 10 mM imidazole, pH 7.8, 300 mM KOAc, 0.2 mM DTT, and 10% [vol/vol] glycerol). The protein was then eluted with buffer C (250 mM imidazole, pH 7.8, 300 mM KOAc, and 10% [vol/vol] glycerol) and stored at −80°C. The protein concentration was measured at $A_{280}$ according to the molar extinction coefficient estimated from the amino acid composition (Pace et al, 1995).

### Monobody solubility test

The concentration of the C4-AM2 monobodies [C4-AM2-(−), C4-AM2-(EIDK), C4-AM2-(EIDKPSQ)] was adjusted to 150 $\mu$M with the buffer B. The buffer was changed to 25 mM Hepes-K, pH 7.5, 150 mM NaCl by gel filtration. UV-vis spectra were measured before (left panel) and after centrifugation (15,300*g*, 5 min; right panel). The protein concentration was calculated from the absorbance at 280 nm of the peak according to the molar extinction coefficient estimated from the amino acid composition.

### Modification of monobodies

Biotin-modified monobodies were prepared as follows: TCEP solution (1 $\mu$l, 500 mM tris[2-carboxyethyl]phosphine, pH 7.6) was added to 100 $\mu$l of the His-tag purified monobody solution, and the resulting solution was incubated at 25°C for 30 min. The quenching solution (3 $\mu$l, 500 mM azidoacetic acid, pH 7.6) was added to the reaction mixture and incubated at 25°C for 30 min. After adding 100 $\mu$l of buffer D (50 mM Hepes-KOH, pH 7.5, 300 mM NaCl, 0.05% [vol/vol] Tween 20, and 0.1% [wt/vol] PEG6000), the buffer was changed to buffer D using Zeba Spin Desalting Columns. The protein concentration in the eluted solution was measured at $A_{280}$. The solution was mixed with 1.5 eq maleimide-PEG11-biotin (Thermo Fisher Scientific). After incubating at 25°C for 30 min, 1.2 eq of DTT was added to the solution, and the solution was stored at −80°C.

Acetamide-modified monobodies were prepared as follows: TCEP solution (1 $\mu$l) was added to 50 $\mu$l of the His-tag purified monobody solution, and the resulting solution was incubated at 25°C for 30 min. The quenching solution (1.5 $\mu$l) was added to the reaction mixture, and the mix was incubated at 25°C for 30 min. Then, 100 mM iodoacetamide (1 $\mu$l) was added to the solution. After 5 min incubation at 25°C, 100 mM DTT (3 $\mu$l) and buffer D (50 $\mu$l) were added, and the resulting solution was incubated at 25°C for 30 min. The buffer in the solution was changed to buffer D using Zeba Spin Desalting Columns, and the protein concentration was measured at $A_{280}$. The solution was stored at −80°C.

### Affinity measurement of monobodies

Affinity measurement was performed on biotinylated monobodies immobilized on a streptavidin biosensor (ForteBio) using the Octet system (ForteBio) as described in the manufacturer's instructions. The analyte RBD was dissolved in water to prepare 15 $\mu$M of RBD solution, and the buffer was changed to buffer D with Zeba Spin Desalting Columns. The protein concentration was measured at $A_{280}$ according to the molar extinction coefficient estimated from the amino acid composition. The RBD solution was stored at −80°C and was used for the following binding assay after dilution with buffer D. The binding assay was performed at 30°C in buffer D. Each step in the binding assay was as follows: equilibration for 150 s, association for 600 s, and dissociation for 600 s.

### Virus neutralization assay

SARS-CoV-2 neutralization assay was performed using VeroE6/TMPRSS2 cells that were obtained from JCRB cell bank. The cells

$(5 \times 10^3$ cells per well, 50 $\mu$l) were seeded into 96-well culture plates and incubated at 37°C for 18 h before infection. SARS-CoV-2 isolate B.1.1 (GISAID# EPI_ISL_568558), a major lineage in 2020 in Japan, was used as a control. The VOC isolates, Alpha variant (B.1.1.7, QK002 strain, GISAID# EPI_ISL_768526), Beta variant (B.1.351, TY8-612 strain, GISAID# EPI_ISL_1123289), Gamma variant (P.1, TY7-503 strain, GISAID# EPI_ISL_877769), and Delta variant (B.1.617.2, TY11-927, GISAID# EPI_ISL_ 2158617) were obtained from the National Institute of Infectious Diseases. Monobodies were serially diluted (from 60 nM to 6 pM) and incubated with an equal volume of the SARS-CoV-2 (4,000 TCID$_{50}$/ml) for 1 h at 37°C. The monobody-virus mixtures (100 $\mu$l) were added to each culture well and incubated for 1 h at 37°C. As a control antibody, anti-SARS-CoV-2 RBD potent neutralizing antibody AM128 (AcroBiosystems) was used in this study. The supernatant was removed, and 100 $\mu$l of fresh Dulbecco's modified Eagle medium (Sigma-Aldrich) supplemented with 10% fetal bovine serum, penicillin (100 U/ml), and streptomycin (100 $\mu$g/ml) (Thermo Fisher Scientific) was added. After incubation in a 37°C-incubator supplied with 5% $CO_2$ for 18 h, the culture supernatants were harvested. The SARS-CoV-2 RNA amounts in the supernatants were measured by RT-qPCR using the PrimeDirect Probe RT-qPCR Mix (Takara Bio) and an N2 primer/probe set (Kondo et al, 2020). The half-maximal inhibitory concentrations ($IC_{50}$) were determined using GraphPad Prism version 9.0.

## Data Availability

All data generated or analyzed during this study are included in this published article and its supplementary information files.

## Supplementary Information

## Acknowledgements

We thank Mr. Hikaru Saiki for helpful discussions during the early phase of this project. This work was supported by AMED (grant numbers 20he0622010h0001 and JP21zf0127004 to H Murakami and 20fk0108293s0101 to Y Iwatani), Grant-in-Aid for Scientific Research on Innovative Areas (Grant Number 20H04704 to G Hayashi), and Grant-in-Aid for Scientific Research (C) (Grant Number 21K05270 to T Fujino) from the Japan Society for the Promotion of Science; and a donation from Dr. Hiroshi Murakami.

### Author Contributions

T Kondo: resources, data curation, formal analysis, validation, investigation, visualization, methodology, and writing—original draft, review and, editing.
K Matsuoka: resources, data curation, formal analysis, validation, investigation, visualization, methodology, and writing—original draft, review, and editing.
S Umemoto: resources, data curation, formal analysis, validation, visualization, and writing—original draft, review, and editing.
T Fujino: funding acquisition and writing—original draft, review, and editing.
G Hayashi: funding acquisition and writing—original draft, review, and editing.
Y Iwatani: conceptualization, resources, formal analysis, supervision, funding acquisition, validation, investigation, visualization, methodology, project administration, and writing—original draft, review, and editing.
H Murakami: conceptualization, resources, formal analysis, supervision, funding acquisition, validation, investigation, visualization, methodology, project administration, and writing—original draft, review, and editing.

### Conflict of Interest Statement

T Kondo, T Fujino, S Umemoto, G Hayashi, Y Iwatani, and H Murakami are inventors on the provisional patent application (PCT/JP2021/018668; filed 5/18/2021) submitted by Tokai National Higher Education and Research System and National Hospital Organization Nagoya Medical Center. The patent application is for monobody sequences against the SARS-CoV-2 spike protein. Other authors declare that they have no competing interests.

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
