## [Reviewer comments · Life Science Alliance]

Life Science Alliance

Monobodies with potent neutralizing activity against SARS-CoV-2 Delta and other variants of concern

Taishi Kondo, Kazuhiro Matsuoka, Shun Umemoto, Tomoshige Fujino, Gosuke Hayashi, Yasumasa Iwatani, and Hiroshi Murakami

DOI: <https://doi.org/10.26508/lsa.202101322>

Corresponding author(s): Hiroshi Murakami, Nagoya University

Review Timeline:

Submission Date:	2021-11-28
Editorial Decision:	2022-01-11
Revision Received:	2022-01-20
Editorial Decision:	2022-02-11
Revision Received:	2022-02-17
Accepted:	2022-02-18

Scientific Editor: Novella Guidi

Transaction Report:

January 11, 2022

Re: Life Science Alliance manuscript #LSA-2021-01322-T

Dr. Hiroshi Murakami
Nagoya University
Department of Biomolecular Engineering, Graduate School of Engineering
Furo-cho
Chikusa-ku
Nagoya, Aichi 464-8603
Japan

Dear Dr. Murakami,

Thank you for submitting your manuscript entitled "Monobodies with potent neutralizing activity against SARS-CoV-2 delta and other variants of concern" to Life Science Alliance. The manuscript was assessed by expert reviewers, whose comments are appended to this letter. We, thus, encourage you to submit a revised version of the manuscript back to LSA that responds to all of the reviewers' points.

Thank you for this interesting contribution to Life Science Alliance. We are looking forward to receiving your revised manuscript.

Sincerely,

B. MANUSCRIPT ORGANIZATION AND FORMATTING:

Reviewer #1 (Comments to the Authors (Required)):

Kondo et al. report affinity maturation of the SARS-CoV-2 spike-binding monoclonal antibodies developed by the authors previously and neutralization activities of these optimized monoclonal antibodies against wild-type SARS-CoV-2 and the recent variants of concern. This is a topic of high general interest. The very high RBD-binding affinities and neutralizing activities of the optimized monoclonal antibodies (C4-AM2 and C12b-AM2 have $K_D < 0.01$ nM) are impressive, making them useful reagents with possible diagnostic or therapeutic potentials. The data shown are of high quality and the presentation is clear and easy to follow. I have questions regarding selectivity of monoclonal antibodies in general. Please see comments below on this and other points.

Why is C12b-AM2 not included in the analyses presented in Figs. 3, 4?

How selective are these monoclonal antibodies, the C6b and C12b series in particular? Could the authors test binding to a negative control protein (something other than RBD) in BLI?

The mixed bases notation (N = A, C, G, T ; K = G or T) should be defined.

"We found that the extension of the natural sequence at the C-terminal improved the monoclonal antibody solubility (data not shown)." This point should be elaborated. "C-terminal" should be C-terminus.

Undetectable k_{off} Undetectable dissociation or unmeasurable k_{off}

K_D was not detected K_D was not determined

In Table 1, what do (DR) and (TS) represent?

"Since the K_D values observed against the RBDs of these variants were 1.7 to 6.3 nM, the IC_{50} values were 10 times higher than we expected. The monoclonal antibody C4-AM2 might have fast and slow dissociation rates (Fig. 3 and S4), and the slow dissociation rate contributed to the low IC_{50} values."

The slow dissociation contributes to high affinity (small K_D). It does not explain the much lower IC_{50} compared to K_D . Is it possible that virus neutralization is not simply due to RBD-binding?

"C4-AM2 might have fast and slow dissociation rates" fast association and slow dissociation?
"10 times higher" lower

Reviewer #2 (Comments to the Authors (Required)):

This manuscript describes the use of the mRNA-based TRAP display system to affinity mature the original monoclonal antibodies previously discovered against the COVID-19 virus Receptor Binding Domain (RBD). The residues in the BC and FG loops were made into saturated combinatorial libraries from which the most frequently found clones by the next generation sequencing techniques were identified for the next round of affinity maturation campaign. The resulting monoclonal antibody variants had up to almost 100-fold increase in affinity against the RBD. The affinity matured monoclonal antibodies also showed increased affinity towards 4 major COVID-19 variants of concern (VOC). Importantly, the improved affinity correlated to enhanced neutralization potency against the wt and VOCs.

The manuscript was described in great technical detail, particularly the Methods section. The techniques used were state of the art and the designs of the experiments were appropriate. The conclusions, both the affinity and the neutralization potency, were strongly supported by the data presented. Overall, this manuscript appears to only present facts of the study but lacks elaboration of why it is interesting and important given that discovery, engineering and characterization of many antibodies,

especially the nanobodies isolated from the llamas that are similar in both the structure and size, have been extensively published in the past year. If the affinity maturation is the focus, saturation mutagenesis is already a standard and well-established protein engineering technique, hence obtaining improved clones, while interesting, is not too surprising. What's surprising is the finding that the monobodies in their monomeric form achieved extremely high neutralizing potency comparable to the bivalent neutralizing IgG, with EC50 values lower than their equilibrium constants (affinity). The authors are encouraged to further elaborate on these findings and speculate the possible reasons. Would the high potency be the result of protein aggregation? Lastly, would the bivalent monobodies in the Fc-fusion form further enhance neutralization potency? Given the rapid clearance of small proteins such as the monobodies, this bivalent format widely used in therapeutic biologics deserves further studies.

Reviewer #3 (Comments to the Authors (Required)):

The authors have published a paper in Science Advances in October 2020 where they use an innovative selection method, termed TRAP display, to engineer high-affinity FN3-monobodies to the RBD of the spike protein of SARS-CoV2. These monobodies bound with low nanomolar affinities to the RBD, but showed only moderate virus neutralization activity. In this study, the authors used affinity maturation to increase binding affinity and neutralization activity of three different monobody clones from their previous study. The resulting monobodies have an impressive affinity in the sub-nanomolar range with significantly slower koff rates and have potent neutralizing activity (up to 100-fold better than parent monobody) against different SARS-CoV2 spike mutants. Monobody C4-AM4 was the most potent clone and showed similar activity than a validated neutralizing monoclonal antibody. The developed monobodies are among the most potent neutralizing binding proteins against SARS-CoV2 and have great promise as therapeutic or diagnostic reagents.

This is a rigorously planned and conducted study with important results that should be published without delay.

Minor points:

1. page 3: Was the improved soluble expression of monobody C4-AM1 with C-terminal extension also observed for the parental C4 monobody?
2. page 3: The authors interpret the differences of C4 affinity in their previous study (1.9 nM) vs. this study (0.96 nM) to the removal of the Nus-Tag. But are these small affinity differences really significant and not within the error margins of SPR?
3. The manuscript would benefit from proofreading to remove typos and grammatical errors.

Response to Reviews:

We are very grateful for the reviewer's comments and suggestions, which we found very helpful. We have revised our manuscript accordingly.

Reviewer #1 comments:

- I have questions regarding selectivity of monobodies in general. Please see comments below on this and other points.

Why is C12b-AM2 not included in the analyses presented in Figs. 3, 4?

Response: In a previous paper (Science Advances, 2020, 6(42), eabd3916.), we showed that C12b did not prevent SARS-CoV-2 infection. In Figure 3, we did not include C12b-AM2 because we focused on analyzing an affinity of a neutralizing monobody. In Figure 4, we have added C12b as a negative control. We did not use the C12b-AM2 monobody because it would not have a neutralizing activity.

How selective are these monobodies, the C6b and C12b series in particular? Could the authors test binding to a negative control protein (something other than RBD) in BLI?

Response: We performed a BLI assay using a green fluorescent protein and a ubiquitin protein as negative control proteins. We observed no binding. Please see the related manuscript file.

The mixed bases notation (N = A, C, G, T ; K = G or T) should be defined.

Response: Thank you for your suggestion. We have added it to the main text and the figure legends.

"We found that the extension of the natural sequence at the C-terminal improved the monobody solubility (data not shown)." This point should be elaborated. "C-terminal" should be C-terminus.

Undetectable koff è Undetectable dissociation or unmeasurable koff

KD was not detected è KD was not determined

"10 times higher" è lower

Response: Thank you for your corrections.

In Table 1, what do (DR) and (TS) represent?

Response: There is a deletion of 2 amino acids (VR) after the random sequence for these clones. We

removed the brackets and have added the following note in the caption of Table 1: “*These clones had a two amino acid (VR) deletion after the BC loop.” We have revised Figs. S3 and S4 as well.

"Since the K_D values observed against the RBDs of these variants were 1.7 to 6.3 nM, the IC_{50} values were 10 times higher than we expected. The monobody C4-AM2 might have fast and slow dissociation rates (Fig. 3 and S4), and the slow dissociation rate contributed to the low IC_{50} values."

"C4-AM2 might have fast and slow dissociation rates" è fast association and slow dissociation?

Response: We would like to draw attention to the unfitted dissociation curve in Fig. 3. To make this sentence clear, we revised it to: “the C4-AM2 monobody might have two dissociation rates (Supplementary Fig. S4)”

The slow dissociation contributes to high affinity (small K_D). It does not explain the much lower IC_{50} compared to K_D . Is it possible that virus neutralization is not simply due to RBD-binding?

Response: I agree; it does not fully explain the results, especially for the Beta and Gamma variants. At the moment, we really do not have other explanations. We have revised the sentence to read, “and the slow dissociation rate might partially contribute to the low IC_{50} values.”

Reviewer #2 comments:

Overall, this manuscript appears to only present facts of the study but lacks elaboration of why it is interesting and important given that discovery, engineering and characterization of many antibodies, especially the nanobodies isolated from the llamas that are similar in both the structure and size, have been extensively published in the past year. If the affinity maturation is the focus, saturation mutagenesis is already a standard and well-established protein engineering technique, hence obtaining improved clones, while interesting, is not too surprising. What's surprising is the finding that the monobodies in their monomeric form achieved extremely high neutralizing potency comparable to the bivalent neutralizing IgG, with EC_{50} values lower than their equilibrium constants (affinity). The authors are encouraged to further elaborate on these findings and speculate the possible reasons. Would the high potency the results of protein aggregation?

Response: We think the extremely high affinity ($K_D < 0.01$ nM) of the monobody C4-AM2 and the potent neutralization activity ($IC_{50} = 46$ pM, 0.62 ng/mL) are interesting. We agree that the IC_{50} values being lower than

their K_D values for VOCs is also interesting. We did not observe protein aggregation of the C4-AM2 monobody in the neutralization experiment. We also tried to determine the K_D values for a spike protein trimer, but it was difficult due to the high non-specific binding of the trimer to the sencer. At the moment, we do not have an answer, although we think the fraction of the slow dissociation rate might partially contribute to the low IC_{50} values.

Lastly, would the bivalent monobodies in the Fc-fusion form further enhance neutralization potency? Given the rapid clearance of small proteins such as the monobodies, this bivalent format widely used in therapeutic biologics deserves further studies.

Response: We never tried the Fc-fusion because of the relatively high cost of production. Instead, we tried a tandem format in the previous work (Science Advances, 2020, 6(42), eabd3916.), but this did not improve the neutralization activity. For an *in vivo* animal test, we will test the Fc-fusion to reduce the clearance rate.

Reviewer #3 comments:

1. page 3: Was the improved soluble expression of monobody C4-AM1 with C-terminal extension also observed for the parental C4 monobody?

Response: Here, we mentioned our general observations of the effects of C-terminal extension of various monobodies on their solubility. To make it clearer, we have added the solubility study of monobody C4-AM2 with three types of C-termini as Fig. S1. We revised the sentence as, “During the optimization of monobody expression, we found that the extension of the natural sequence at the C-terminus improved monobody solubility (Supplementary Fig. S1; monobody C4-AM2, *vide infra*).

2. page 3: The authors interpret the differences of C4 affinity in their previous study (1.9 nM) vs. this study (0.96 nM) to the removal of the Nus-Tag. But are these small affinity differences really significant and not within the error margins of SPR?

Response: We agree with this comment. We removed this sentence in the revised version.

3. The manuscript would benefit from proofreading to remove typos and grammatical errors.

Response: According to the reviewer’s suggestion, we have removed typos and grammatical errors from the manuscript through proofreading.

February 11, 2022

RE: Life Science Alliance Manuscript #LSA-2021-01322-TR

Dr. Hiroshi Murakami
Nagoya University
Department of Biomolecular Engineering, Graduate School of Engineering
Furo-cho
Chikusa-ku
Nagoya, Aichi 464-8603
Japan

Dear Dr. Murakami,

Thank you for submitting your revised manuscript entitled "Monobodies with potent neutralizing activity against SARS-CoV-2 Delta and other variants of concern". We would be happy to publish your paper in Life Science Alliance pending final revisions necessary to meet our formatting guidelines.

- please address the remaining Reviewer 1 point
- please add the Twitter handle of your host institute/organization as well as your own or/and one of the authors in our system
- please add callouts for Figures S2A-B; S3A-C and S4A-B to your main manuscript text;
- please provide a separate Data Availability section

A. FINAL FILES:

B. MANUSCRIPT ORGANIZATION AND FORMATTING:

Sincerely,

Reviewer #1 (Comments to the Authors (Required)):

The authors have adequately addressed all questions/comments from the previous review.
One small point is regarding the last sentence of the Results section; the BLI data for C4-AM2 are not in supplementary Fig S4.

Reviewer #2 (Comments to the Authors (Required)):

The authors have sufficiently addressed the questions I raised in their responses. I look forward to the in vivo data using the half-life enhanced molecular format, such as the Fc-fusion.

Reviewer #3 (Comments to the Authors (Required)):

The authors fully addressed all points raised in my initial review. No further comments.

February 18, 2022

RE: Life Science Alliance Manuscript #LSA-2021-01322-TRR

Dr. Hiroshi Murakami
Nagoya University
Department of Biomolecular Engineering, Graduate School of Engineering
Furo-cho
Chikusa-ku
Nagoya, Aichi 464-8603
Japan

Dear Dr. Murakami,

Thank you for submitting your Research Article entitled "Monobodies with potent neutralizing activity against SARS-CoV-2 Delta and other variants of concern". It is a pleasure to let you know that your manuscript is now accepted for publication in Life Science Alliance. Congratulations on this interesting work.

DISTRIBUTION OF MATERIALS:

Again, congratulations on a very nice paper. I hope you found the review process to be constructive and are pleased with how the manuscript was handled editorially. We look forward to future exciting submissions from your lab.

Sincerely,
